# Telling the Wood from the Trees: Ranking a Tree Species List to Aid Urban Afforestation in the Amazon

**Daniela C. Zappi** [1,2,*], **Juliana Lovo** [3], **Alice Hiura** [4], **Caroline O. Andrino** [4], **Rafael G. Barbosa-Silva** [2,4], **Felipe Martello** [4], **Livia Gadelha-Silva** [2,5], **Pedro L. Viana** [5] and **Tereza C. Giannini** [4]

1. Programa de Pós-Graduação em Botânica, Instituto de Ciências Biológicas, Universidade de Brasília, Caixa Postal 04457, Brasília 70919-970, DF, Brazil
2. UFRA/MPEG Programa de Pós-Graduação em Ciências Biológicas, Museu Paraense Emílio Goeldi, Coord. Botânica, Belém 66077-830, PA, Brazil; rafa.g29@gmail.com (R.G.B.-S.); liviaagadelha@gmail.com (L.G.-S.)
3. Departamento de Sistemática e Ecologia, Universidade Federal da Paraíba, Caixa Postal 5065, Cidade Universitária, João Pessoa 58051-970, PB, Brazil; juliana.lovo@dse.ufpb.br
4. Instituto Tecnológico Vale, Belém 66055-090, PA, Brazil; alicehiura@gmail.com (A.H.); caroline.andrino@pq.itv.org (C.O.A.); felipemartello@gmail.com (F.M.); tereza.giannini@itv.org (T.C.G.)
5. Museu Paraense Emílio Goeldi, Coord. Botânica, Belém 66077-830, PA, Brazil; pedroviana@museu-goeldi.br
* Correspondence: danielazappi14@gmail.com

**Abstract:** The vast Amazonian biome still poses challenges for botanists seeking to know and recognize its plant diversity. Brazilian northern cities are expanding fast, without considering the regional biodiversity, and urban plantings of almost exclusively exotic species are taking place. It is paramount that the correct identity of such trees is ascertained before procurement of the seeds and young plants, as the use of popular names may lead to importation of plant material from elsewhere, with potential introduction of invasive species. The abundant local diversity also leads to the need to score the most suitable species within a given region. Following the preparation of authoritatively named floristic lists in Southeastern Pará state, we proceeded to score and rank the most suitable trees for urban planning using different characteristics such as size, ornamental value, ecologic role, resilience and known methods of propagation. From an initial 375 species list, 263 species were ranked according to their suitability for street and urban area plantings and visualized using a Venn diagram. A final list with the 49 of the highest-ranking species was further analysed regarding their pollination and phenology period and two types of dissimilarity analyses were provided to aid practitioners in matching and choosing groups of species. Different local vegetation types mean that similar floristic lists must be used to extract cohorts of suitable plants to increase the urban richness in the eight Brazilian states that are included in the Amazonian biome.

**Keywords:** Aichi targets; Carajás; floristic list; Principal Coordinates Analysis; selecting species; tree planting

## 1. Introduction

The vast Amazonian biome still poses challenges for botanists seeking to know and recognize its plant diversity [1,2]. Paradoxically, protecting Amazonian rainforests seems to be hampered by its own high biodiversity, as the sheer extent of the biome makes it very difficult to select adequate species for local reforestation [3]. Brazilian northern cities are expanding fast, changing local microclimate [4,5] and without taking into account the biodiversity of the region, as urban plantings of almost exclusively exotic species take place [5–7]. It is paramount that the correct identity of the trees is ascertained before procurement of seeds and young plants, as the use of popular names may lead to the importation of plant material from Brazilian southern states and elsewhere, with the potential introduction of inadequate and even invasive species [8–10].

The abundant local plant diversity leads to the need to score the most suitable species within a given region [3]. While Brasília, the capital of Brazil, is an excellent example of

urban planning that includes the marked presence of native and ornamental trees lining its streets and parks [11], the situation in a large number of cities in the Northeast [12–15] and North [16–18] regions of Brazil is much less encouraging. While in some cities there are examples of the introduction of Brazilian native trees, the actual dominance of exotic species tends to invalidate these positive initiatives [19]. Two Amazonian capitals, Belém and Manaus, have the lowest indices of urban afforestation in the region, and the majority of species used is exotic [5].

Proven benefits from urban afforestation are not only direct consequences for locals [20] but also for the biodiversity, climate and ecological functions that, at the same time, have an indirect impact on the local human population. Urban trees have a major role in the re-establishment of wildlife, positively impacting the bird fauna [21] and improving and controlling the microclimate [22–24]. Additionally, the importance of these urban forests has been highlighted in events such as the COVID-19 pandemic, which has seen an increased demand from society to access green areas within towns and cities [25,26]. In fact, the reasons for conserving urban biodiversity are many-fold, including preservation of local biodiversity, stepping stones connecting urban and non-urban habitats, responding to environmental change, providing ecosystem services and material for environmental education, as well as ethical and human well-being reasons [27]. Street and park plantings represent the foundations for a more comprehensive approach, and the choice of species is intimately linked with the good functioning of the wider network. International targets, such as the Aichi Biodiversity Targets [28], aim to ensure urban greening (including tree planting) is legally protected on a country-wide basis. The most relevant targets (Target 14 and 15) are linked to restoration of essential ecosystem services that contribute to health, livelihoods and well-being, besides contributing to mitigation of climate change and restoration of degraded ecosystems [29].

The present work takes advantage of the preparation of authoritatively named floristic lists for the forests in the Carajás area, southeastern Pará state as a robust source of information that will serve as a basis to score and rank the most suitable trees for urban afforestation in the growing cities of the region of the Itacaiúnas River basin [30] (Parauapebas, Canaã dos Carajás, Marabá, Ourilândia do Norte and Tucumã—Figure 1), using differential characteristics such as size, ornamental value, ecological role, resilience and known methods of propagation.

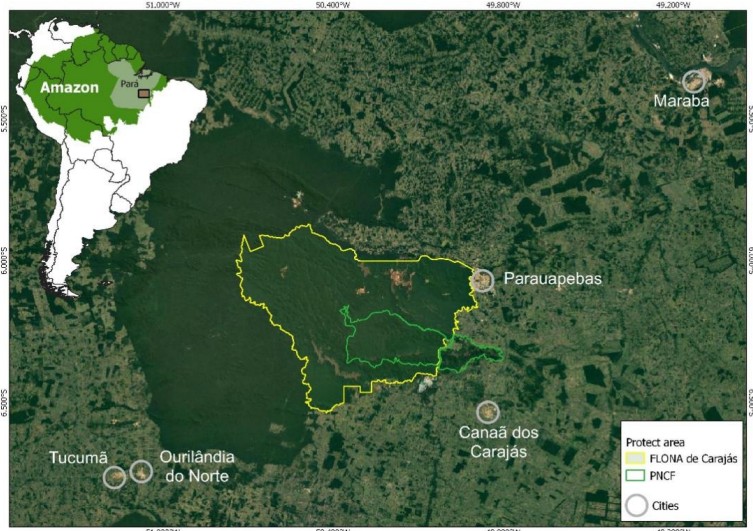

**Figure 1.** Map of the region showing towns of Canaã dos Carajás, Ourilândia do Norte, Parauapebas, Tucumã and Marabá (grey circles) in relation to the area where the survey was carried out (FLONA de Carajás—yellow outline and Parque Nacional dos Campos Ferruginosos—PNCF—green outline). Dark green cover represents pristine primary vegetation, mottled grey and green denotes deforestation in the region.

## 2. Material and Methods

Tree species were extracted from a local checklist of the forest vegetation of the Floresta Nacional de Carajás (FLONA Carajás) and the Parque Nacional dos Campos Ferruginosos (PNCF) (Figure 1) produced by our team between 2016 and 2019 [31]. All species recorded in this list have vouchers deposited at the herbarium of the Museu Paraense Emílio Goeldi (MG) and/or at the Herbário de Carajás (HCJS).

From an initial list of 375 species (Table S1), we eliminated all taxa that were not native from Pará or that were not confirmed as belonging to a certain species. We also eliminated plants that had variable habit through double checking in databases, such as Flora do Brasil 2020 [32] and virtual herbaria [33,34], and removed taxa that had lianescent or shrubby habit. A similar procedure was used to detect and exclude species with large tabular or buttress roots, plants with spiny stems that might be considered hazardous in an urban setting and species that are known to be toxic or invasive. These criteria were based upon the authors' experience and also followed guidelines suggested by similar investigations and guides for urban afforestation (e.g., [35,36])

Ranking of the remaining tree species was based on tree size, ornamental value, ecological impact, Amazonian distribution, propagation methods and resilience, as explained in Table 1. During this first phase of analysis all six variables considered had the same weight, and not one was considered more important than another.

**Table 1.** Tree species score according to the six important features for urban greening, where 'yes' was attributed for ideal/adequate situation and 'no' when species did not meet the requirement.

| | Feature | Ideal/Adequate |
|---|---|---|
| A | Appropriate height | Species that reach up to 10–15 m tall were preferred over taller species for street planting, however taller species were not eliminated from our list. |
| B | Showy flowers | It has been established that the population is keen on trees that produce ornamental displays in different seasons. |
| C | Fruits attractive to wildlife and to humans | With the intention of including more ecological interactions in the urban spaces, species with fruits that are attractive to wildlife were given priority. Moreover, streets planted with fruiting trees are a huge bonus to develop links between people and plants from an early age. |
| D | Amazonian distribution | Trees that are endemic to the Amazon were given preference as the list was prepared for urban areas within this biome. |
| E | Known propagation methods | Scientific and horticultural knowledge were combined to score this feature. We have considered that, if the propagation of a genus is known, this knowledge is transferrable to the species. |
| F | Resilience (drought resistance and/or tolerance to full sun) | Even in the extremely humid Amazonian climate, urban weather conditions differ from the natural situation. Here the intention was to favour species adapted to growing at the edge of forests, in more seasonal forest types or in open vegetation in the Amazon to ensure that these can thrive in urban conditions. |

The features of each species were analysed by checking against information from the literature (floristic accounts, taxonomic revisions when available), online data repositories [32–34] and our field knowledge of individual species. The resulting ranked tree species were presented using InteractiVenn [37], to enable users to visualize the list of species that fulfilled different features and, through overlap of these lists, to select species that are most apt for urban afforestation.

A list of selected species that fulfilled all or five out of six features was presented together with their pollination syndrome [38] and phenology data retrieved from relevant bibliography, such as floristic accounts and family revisions for individual taxa, and, in their absence, virtual herbarium databases [33,34].

We also investigated the complementarity between the 49 selected species, aiming to guide urban afforestation strategies that seek to provide greater diversity and sustainability of resources. For this, we consider as species complementarity the dissimilarity between species based on three features: (i) pollination syndrome, (ii) showiness and (iii) flowering time. We consider that the first two features indicate the diversity of characteristics of the trees and the last refers to the availability of resources throughout the year. Tree species were classified according to pollination syndrome considering eight pollination agents: beetles, bees, butterflies, birds, bats, moths, flies, and insects (this last category considers pollination syndromes that do not obviate a group of insects or that might attract more than one insect group). The tree species were also classified regarding the appearance of their flowers as showy or not showy. Considering the flowering of the species, the phenology data assembled in Table 2 was used.

**Table 2.** 49 tree species that score positively for 6(*-7 top species) or 5 criteria, with pollination and phenology information (Fl = flower, Fr = fruit; I-January, II-February, III-March, IV-April, V-May, VI-June, VII-July, VIII-August, IX-September, X-October, XI-November, XII-December, AY-all year long).

| | Family | Species | Criteria | Pollination Syndrome | Phenology |
|---|---|---|---|---|---|
| 1 | Annonaceae | *Annona exsucca* DC. | A, C–F | Beetle | Fl XI–V, Fr III–VI |
| 2 | Annonaceae | *Xylopia aromatica* (Lam.) Mart. | A–C, E, F | Beetle | Fl VIII–V, Fr III–VIII |
| 3 | Apocynaceae | *Tabernaemontana flavicans* Willd. ex Roem. & Schult. | A–C, E, F | Bee | Fl VIII–X, Fr IX–VII |
| 4 | Bixaceae | *Cochlospermum orinocense* (Kunth) Steud. | A–C, E, F | Bee | Fl IV–X, Fr VI–IV |
| 5 | Boraginaceae | *Cordia goeldiana* Huber | A, B, D–F | Bee, Hoverfly | Fl VI–X, Fr IX–XII |
| 6 | Burseraceae | *Protium sagotianum* Marchand | A, C–F | Bee | Fl VIII–I, Fr XI–VI |
| 7 | Chrysobalanaceae | *Hirtella racemosa* Lam. | A, B, D–F | Butterfly | Fl II–VII, Fr VIII–X |
| 8 | Clusiaceae | *Clusia panapanari* (Aubl.) Choisy | A–C, E, F | Bee | Fl II–XI, Fr IX–I |
| 9 | Clusiaceae | *Symphonia globulifera* L.f. | A–C, E, F | Bird, Hummingbird | Fl VII–II, VI–V, Fr VIII–II |
| 10 | Ebenaceae | *Diospyros vestita* Benoist | A, C–F | Insect | Fl VII–II, Fr VI–VII |
| 11 | Fabaceae | *Abarema cochleata* (Willd.) Barneby & J.W.Grimes | A, B, D–F | Bat, Moth | Fl VIII–I, Fr AY |
| 12 | Fabaceae | *Bauhinia acreana* Harms | A, B, D–F | Bat | Fl X–V, Fr XI–VI |
| 13 | Fabaceae | *Bowdichia nitida* Spruce ex Benth. | A, B, D–F | Bee | Fl IV–VII, Fr VI–IX |
| 14 | Fabaceae | *Campsiandra laurifolia* Benth. | A, B, D–F | Bat | Fl V–I, Fr VI–IV |
| 15 | Fabaceae | *Cassia fastuosa* Willd. ex Benth. | A, B, D–F | Bee | Fl VII–XI, Fr XI–VIII |
| 16 | Fabaceae | *Cenostigma tocantinum* Ducke | A, B, D–F | Bee | Fl IV–II, Fr V–XII |
| 17 | Fabaceae | *Hymenaea parvifolia* Huber | B–F | Bat | Fl XI–III, Fr AY |
| 18 | Fabaceae | *Inga disticha* Benth. | A–C, E, F | Bee, Butterfly | Fl VI–IX, Fr XI–VII |
| 19 | Fabaceae | *Inga heterophylla* Willd. | A–C, E, F | Bee | Fl XII–VI, IX, Fr V–XI, I |
| 20 | Fabaceae | *Inga marginata* Willd. | A–C, E, F | Bee | Fl VII–XII, Fr IX–IV |
| 21 | Fabaceae | *Inga nobilis* Willd. | A–F | Bat | Fl X–V, Fr II–VI |
| 22 | Fabaceae | *Inga stipularis* DC. | A–E | Moth, Bat | Fl II–VI, XI, Fr I–VII |
| 23 | Fabaceae | *Inga umbellifera* (Vahl) DC. | A–F | Moth | Fl V–IX, Fr VI–X |
| 24 | Fabaceae | *Platymiscium filipes* Benth. | A, B, D–F | Bee | Fl III–XI, Fr VII–III |
| 25 | Fabaceae | *Senna multijuga* (Rich.) H.S.Irwin & Barneby | A, B, D–F | Bee | Fl IV–XII, Fr VII–XII |
| 26 | Fabaceae | *Swartzia arumateuana* (R. S. Cowan) Torke & Mansano | A–F | Bee | Fl II–V, Fr IV–XII |
| 27 | Fabaceae | *Swartzia brachyrachis* Harms | A–F | Bee | Fl III–VI, Fr VI–II |
| 28 | Fabaceae | *Tachigali paniculata* Aubl. | A, B, D–F | Bee | Fr VI–XI, Fr VIII–II |
| 29 | Fabaceae | *Zollernia paraensis* Huber | A–C, E, F | Bee | Fl VIII–X, Fr XI–XII |
| 30 | Lecythidaceae | *Eschweilera obversa* (O.Berg) Miers | B–F | Bee | Fl V–VI, XI–I, Fr V–VI, XII |

**Table 2.** *Cont.*

| | Family | Species | Criteria | Pollination Syndrome | Phenology |
|---|---|---|---|---|---|
| 31 | Malpighiaceae | *Byrsonima spicata* (Cav.) DC. | A–C, E, F | Bee | Fl VII–IV, Fr X–V, VII |
| 32 | Malpighiaceae | *Byrsonima stipulacea* A.Juss. | A–C, E, F | Bee | Fl X–I, Fr V–VII |
| 33 | Malpighiaceae | *Lophanthera lactescens* Ducke | A, B, D–F | Bee | Fl V, X, Fr V |
| 34 | Malvaceae | *Matisia ochrocalyx* K.Schum. | A, B, D–F | Bat | Fl I–VIII, Fr VIII–X, III |
| 35 | Malvaceae | *Theobroma speciosum* Willd. ex Spreng. | A–F | Fly | Fl VIII–XI, Fr X–II |
| 36 | Melastomataceae | *Bellucia grossularioides* (L.) Triana | A–F | Bee | Fl V–II, Fr VIII–IV |
| 37 | Melastomataceae | *Bellucia mespiloides* (Miq.) Macbr. | B–F | Bee | Fl IX–II, VII, Fr X–II |
| 38 | Melastomataceae | *Miconia ampla* Triana | A, C–F | Bee | Fl I–IV, Fr III–VII |
| 39 | Melastomataceae | *Miconia egensis* Cogn. | A, C–F | Bee | Fl III, IX–XI, Fr III, X–XI |
| 40 | Melastomataceae | *Mouriri acutiflora* Naudin | A–F | Bee | Fl VIII–XII, Fr X–II |
| 41 | Melastomataceae | *Mouriri brachyanthera* Ducke | A, C–F | Bee | Fl I, IV–V, XII, Fr I, V–VII |
| 42 | Melastomataceae | *Mouriri cearensis* Huber | A–C, E, F | Bee | Fl X–XII, Fr IV–V, X, XII |
| 43 | Melastomataceae | *Mouriri vernicosa* Naudin | A–E | Bee | Fl V–VIII, X–XII, Fr VII–XI |
| 44 | Myrtaceae | *Eugenia anastomosans* DC. | A, C–F | Bee, Fly | Fl X–XII, Fr VII |
| 45 | Myrtaceae | *Eugenia ramiflora* Desv. ex Ham. | A, C–F | Bee | Fl X–XII, Fr I |
| 46 | Myrtaceae | *Myrcia pyrifolia* (Desv. ex Ham.) Nied. | A, C–F | Bee, Fly | Fl X–I, Fr III–VI, VIII, XII |
| 47 | Myrtaceae | *Psidium acutangulum* DC. | A, C–F | Bee | Fl VI–X, F. IV–VI, IX |
| 48 | Verbenaceae | *Citharexylum macrochlamys* Pittier | A, C–F | Moth | Fl I–IV, VIII, Fr I–IV, VIII |
| 49 | Vochysiaceae | *Erisma uncinatum* Warm. | B–F | Bee | Fl VIII–XI, Fr VII–I |

To calculate the dissimilarity between species, regarding the three characteristics exposed above, we used the Generalized Gower Distance [39], where the pollination syndrome and the flowering time were considered as fuzzy variables and the showiness of the flowers as a variable binary. For each of these variables, the dissimilarity between species and a weighted average of these dissimilarities were calculated, giving three times greater weight to the variables of pollination syndrome and flowering time, than to the variable flower showiness. The different weights for variables are both justified by the importance that we believe each variable has to promote diversity and by the number of categories of these variables. The final range of dissimilarity varies from 0 to 1, where higher values indicated pairs of species with higher complementarity.

The dissimilarity between species was then graphically exposed in two ways, considering two possible approaches to guide urban afforestation management strategies. First, a "tile" type graph was constructed to represent the dissimilarity between pairs of species, enabling managers to identify pairs of more or less complementary species for planting in a given area. Additionally, based on the calculated dissimilarity, we performed a Principal Coordinates Analysis (PCoA), enabling managers to identify a set of species with greater complementarity. In the PCoA figure we also identified the species that had edible and inedible fruits as another possibility to manage urban afforestation strategies. We chose to display this variable (edibility) graphically in the PCoA due to its importance for biodiversity management but did not include it in the analysis. The dissimilarity analysis was performed in R (R Core Team [40] environment using "ade4" package [41] to calculate the generalized Gower distances and "vegan" package [42] to perform the PCoA.

## 3. Results

Twenty-six species that were not native from Pará or that were not confirmed as belonging to a certain species, 67 with variable habit (shrubs or even lianas rather than trees), eight with large tabular or buttress roots, four with spiny stems, three known to be toxic and four possibly invasive trees were eliminated from the initial list.

From the resulting list of 263 species, only seven species scored positively for all six features chosen: *Inga nobilis, Inga umbellifera, Swartzia arumateuana, Swartzia brachyrachis*

(Fabaceae), *Theobroma speciosum* (Malvaceae), *Bellucia mespiloides, Mouriri acutiflora* (Melastomataceae) (see Figure 2), while a second group of 42 species scored positively in five of the six chosen criteria. All of these 49 species are presented in more detail in Table 2, where information on their pollination syndromes and phenology is also displayed.

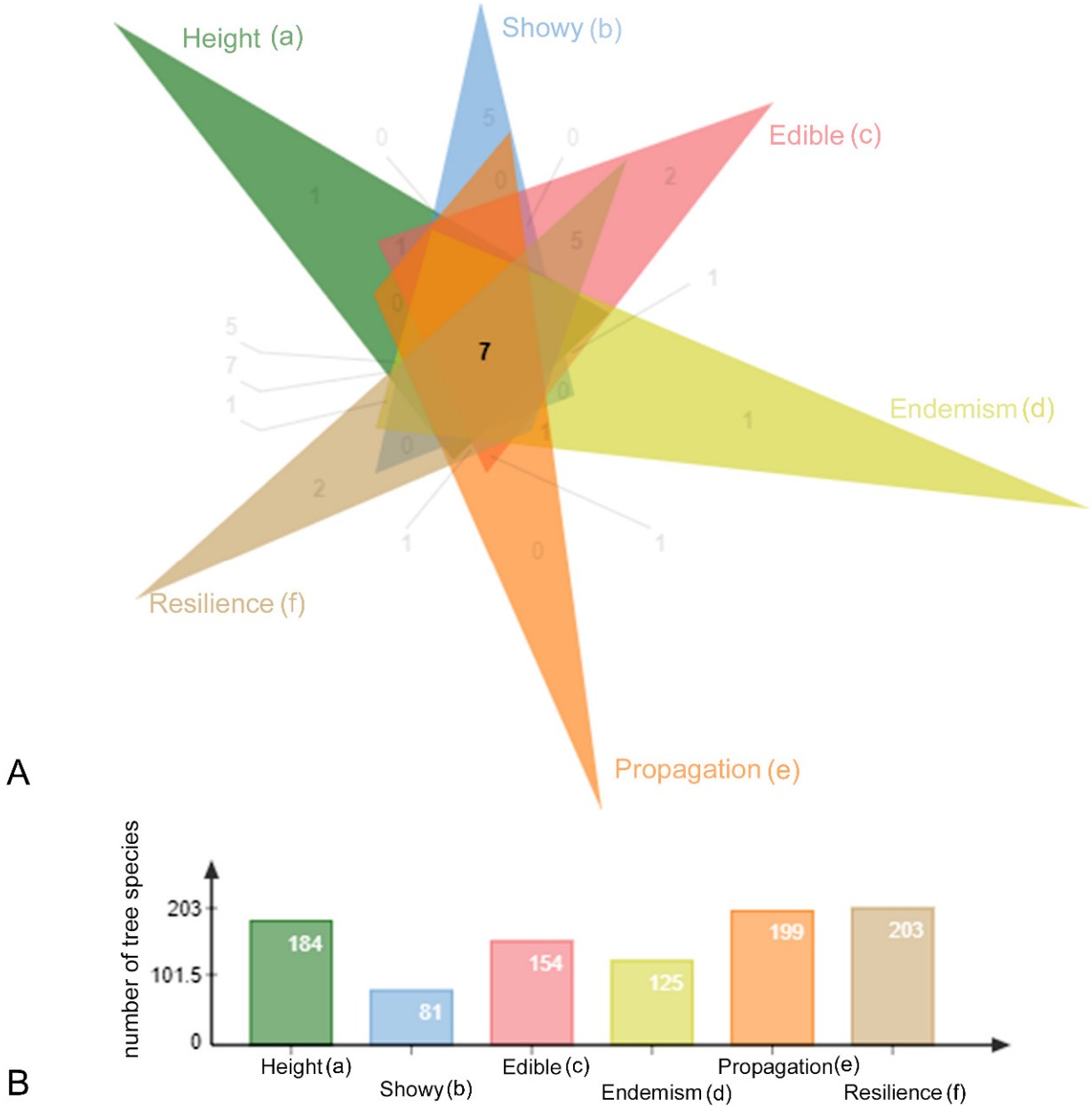

**Figure 2.** (**A**) Venn diagram showing the selected species according to the six chosen features. (**B**) Graph showing the number of tree species grouped per feature. (a. Appropriate height, b. Showy flowers, c. Fruits attractive to wildlife and to humans, d. Amazonian distribution, e. Known propagation methods, f. Resilience).

The criterion that excludes the largest number of the scored species is the ornamental value (69.2%), The second criterion to exclude more species was related to the geographic distribution of the species (52.5%). Species without edible fruits were 41.4% of the species. The height excluded 30% of the species while the lack of propagation knowledge led us to score negatively only 24.3% of the species. Finally, resilience scored negatively for only 22.8% of the studied species.

If considering trees for urban parks and open areas, it is appropriate to use a list that does not take into account the height limit criterion. In this case, *Hymenaea parvifolia* (Fabaceae), *Eschweilera obversa* (Lecythidaceae) and *Ersima uncinatum* (Vochysiaceae) can be added to the list.

*Portraits of Selected Species*

Below, we present portraits and further data of the seven species of tree selected through the scoring system and fulfilling positively all six criteria.

*Inga nobilis (Fabaceae)*

Popular name: ingá, ingá-canela

Trees with compound, alternate paripinnate leaves, rachis winged, sessile, patelliform glands between leaflets, 1–2 pairs of leaflets, flowers arranged in capitate inflorescences, radial corolla fused, stamens showy, white. Indehiscent, leathery pods to 8 × 2.5–3 cm, 5–9 seeds surrounded by white, fleshy, sweet aril.

Flowering from October to May, floral anthesis occurs in the early evening. Bat pollination syndrome, however, hummingbirds have been observed visiting the flowers.

Fruiting February to June, fruit sought after by the local population and dispersed by small mammals, especially monkeys.

Habitat and ecological requirements: understorey species often found in *várzea* forest [43].

Ecological role: associated with nitrogen fixing bacteria, *Inga* species improve soil conditions prior to the introduction of other plantings. This species is frequently used in degraded area recovery projects as it can cope with nutrient poor, acid soils [44].

Propagation (for this genus): Inga seeds are recalcitrant, thus cannot be stored and have to be sown fresh. Remove all pulp and sow in containers in partial shade. Germination occurs within 20–30 days [45].

*Inga umbellifera (Fabaceae)*

Popular name: ingá-xixi-branco

Trees with compound, alternate paripinnate leaves, rachis not winged, sessile, patelliform glands between leaflets, 2–5 pairs of leaflets, flowers arranged in umbellate inflorescences, radial corolla fused, stamens showy, white. Indehiscent, woody pods to 7 × 3 cm, 3–7 rounded seeds surrounded by white, fleshy, sweet aril.

Flowering from May to September, apparently visited by bees, butterflies and other insects.

Fruiting June to November, fruit sought after by the local population and dispersed by small mammals, especially monkeys.

Habitat and ecological requirements: commonly found in riversides this species is also seen in secondary succession areas [46,47]

Ecological role: Due to their nitrogen fixing bacteria, the species of *Inga* are frequently used in agroforestry systems [48], *Inga* species have secondary compounds in their leaves that make them less prone to insect attacks [49]

Propagation: See above species.

*Swartzia arumateuana (Fabaceae)*

Popular name: banha-de-galinha

Trees with compound, alternate imparipinnate leaves, rachis not winged, 8–13 pairs of leaflets, bilateral flowers arranged in racemes, corolla with a single, white petal, stamens showy, in two groups, anthers bright golden-yellow. Indehiscent, hard pods to 6–18 × 3 cm, 6–8 seeds surrounded by yellow, fleshy aril (Figure 3E).

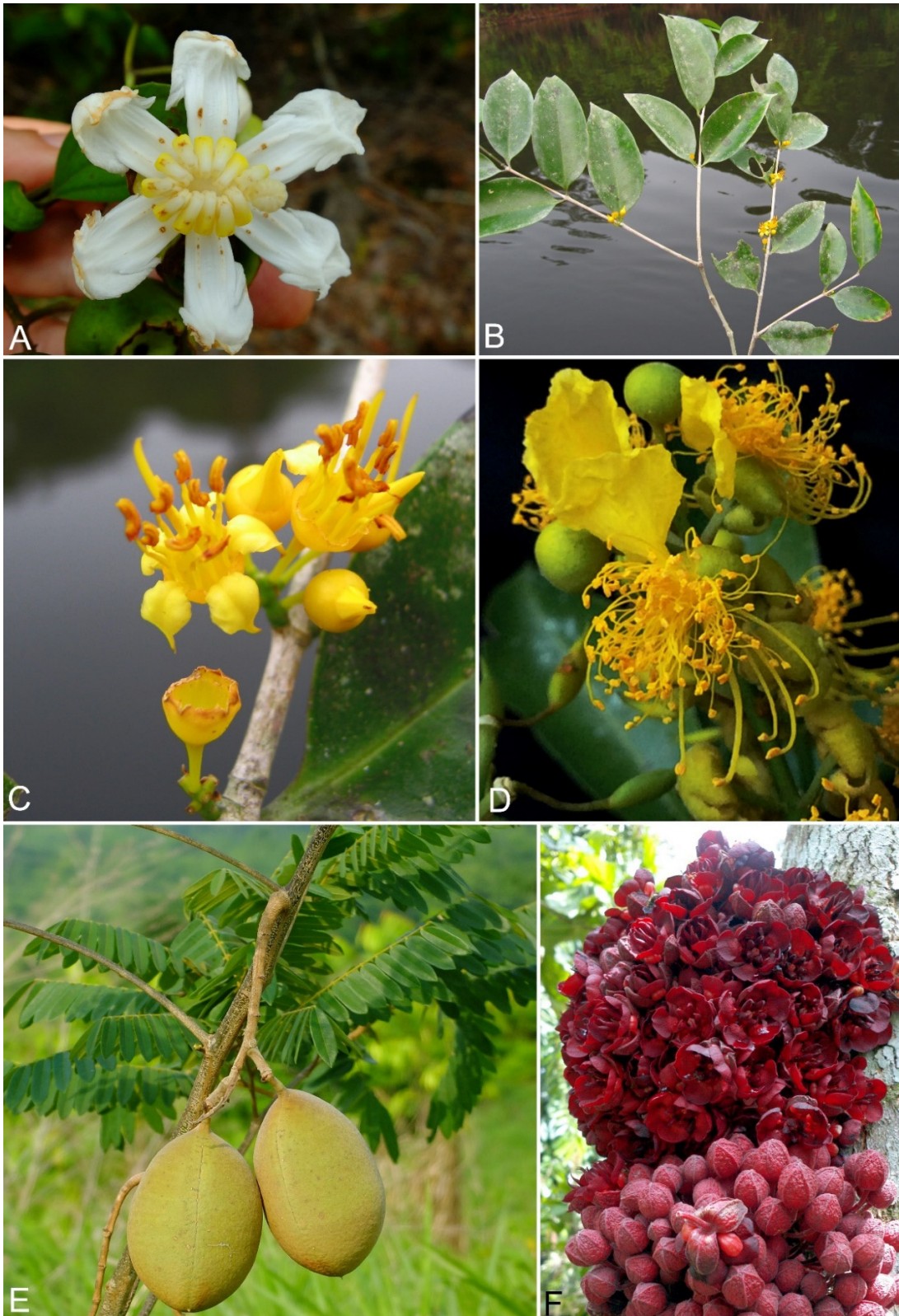

**Figure 3.** Images of tree species selected through the scoring system and fulfilling positively all six criteria. (**A**) Flower of *Bellucia grossularioides*, (**B**,**C**) *Mouriri acutiflora*, habit and flower, (**D**) Flowers of *Swartzia brachyrhachis*, (**E**) Fruit of *Swartzia arumateuana*, (**F**) Cauliflorous *Theobroma speciosum* ((**D**,**F**) by courtesy of L.O.A. Teixeira).

Flowering from February to May, visited by bees and other insects.

Fruiting April to December, fruit sought after by small mammals, especially ground dwelling rodents.

Habitat and ecological requirements: Grows in *terra-firme* forest in clay or sandy soils, and tolerates the presence of iron and manganese [50].

Ecological role: Little is known about the successional preferences of this species, however it was observed flowering and fruiting in open, disturbed fields at the PNCF.

Propagation (for this genus): Seed—best sown as soon as it is ripe in individual containers in a partially shaded position. Germination rates of almost 100% can be expected, with the seed sprouting within 40–50 days [51].

*Swartzia brachyrachis (Fabaceae)*

Popular name: pau-sangue

Trees with simple (unifoliolate) to rarely trifoliolate, alternate leaves, bilateral flowers arranged in racemes, corolla with a single, yellow petal, stamens showy, in two groups, anthers bright golden-yellow. Dehiscent, ovoid pods to 2–3.5 cm, orange when ripe, one-seeded, seed partly surrounded by white aril (Figure 3D).

Flowering from March to June, visited by bees and other insects.

Fruiting June to February, fruit sought after by birds attracted by the aril.

Habitat and ecological requirements: Found in *terra-firme* forest, reaching the canopy [52].

Ecological role: Climax species in dense forest.

Propagation: see above species.

*Theobroma speciosum (Malvaceae)*

Popular name: cacauí

Trees with simple, alternate leaves on branches parallel to the ground, leaves asymmetric, trinerveous at base, much paler beneath, cauliflorous, radial flowers arranged in fascicles and sometimes covering the trunk, petals 5, blood-red. Indehiscent, fleshy, oblong to ellipsoid fruits to $10 \times 5$ cm, pale green turning yellow when ripe, slightly ridged, many seeds surrounded by white, fleshy, sweet aril (Figure 3F).

Flowering from August to November, apparently visited by flies and other insects.

Fruiting October to February, the fruit is a wild relative of cocoa and is sought after by the local population and dispersed by mammals, especially monkeys and rodents [53].

Habitat and ecological requirements: Found in *terra-firme* forest, in the understorey, rarely reaching the canopy [54].

Ecological role: Climax species in dense forest, tolerates shading [54,55].

Propagation: Seed is best sown as soon as fruits ripen in individual containers in a partially shaded position, at temperatures between 20 and 30 °C. Germination takes place within 30 days from sowing.

*Bellucia grossularioides (Melastomataceae)*

Popular name: goiaba de anta

Trees with simple, opposite, curvinerveous leaves, cauliflorous, radial flowers arranged in fascicles along the trunk and branches, petals generally 6, white, stamens yellow, opening by pores. Fleshy round fruits truncate at apex, to 4 cm diam., pale yellow when ripe, many very small seeds imbedded in creamy, sweet pulp (Figure 3A).

Flowering from May to February, apparently visited by large bees.

Fruiting August to April, the fruit is often compared to a guava and is eaten by the local population and dispersed by birds and mammals.

Habitat and ecological requirements: Found in *terra-firme* forest, needs light and abundant water supply to thrive [54].

Ecological role: evergreen species that produces large yields of seeds, is pollinated by bees that are important for Amazonian bee-keeping [54,56]

Propagation: Irrigate water suspended ripe fruit pulp containing seeds on a semi-shaded planted bed filled with organic/sandy substrate. Do not cover seeds, only water

abundantly with fine, gentle spray to ensure the minuscule seeds are buried. Seedlings emerge in few days and germination rate is moderate [57].

*Mouriri acutiflora (Melastomataceae)*

Popular name: araçá-da-várzea

Trees with simple, opposite, glossy and smooth leaves, radial flowers arranged in fascicles along the branches, petals 5, acute, bright yellow, anthers orange-yellow. Fleshy round fruits to 2 cm diam., yellow when ripe, bearing a single seed imbedded in a juicy aril (Figure 3B,C).

Flowering from August to December, sweetly scented and visited by bees.

Fruiting October to February, the fruit is eaten by the local population and dispersed by birds.

Habitat and ecological requirements: Found in *terra-firme*, *igapó* and *várzea* forests, as an understorey tree or near river margins [58].

Ecological role: Climax species pollinated by specialized bees [59].

Propagation (for the genus): Clean seeds benefit from being soaked in giberelic acid in a substrate washed in potassium nitrate [60].

Regarding the complementarity among pair of tree species, values of dissimilarities varied from 0 to 1, with mean of 0.553 and median of 0.603 (Figure 4 and Supplementary Material Table S3). In the PCoA analisys he species that are ordinate closer in the space present smaller dissimilarity values than species ordinate further apart, and this analysis explained 41.39% of species dissimilarities (Figure 5).

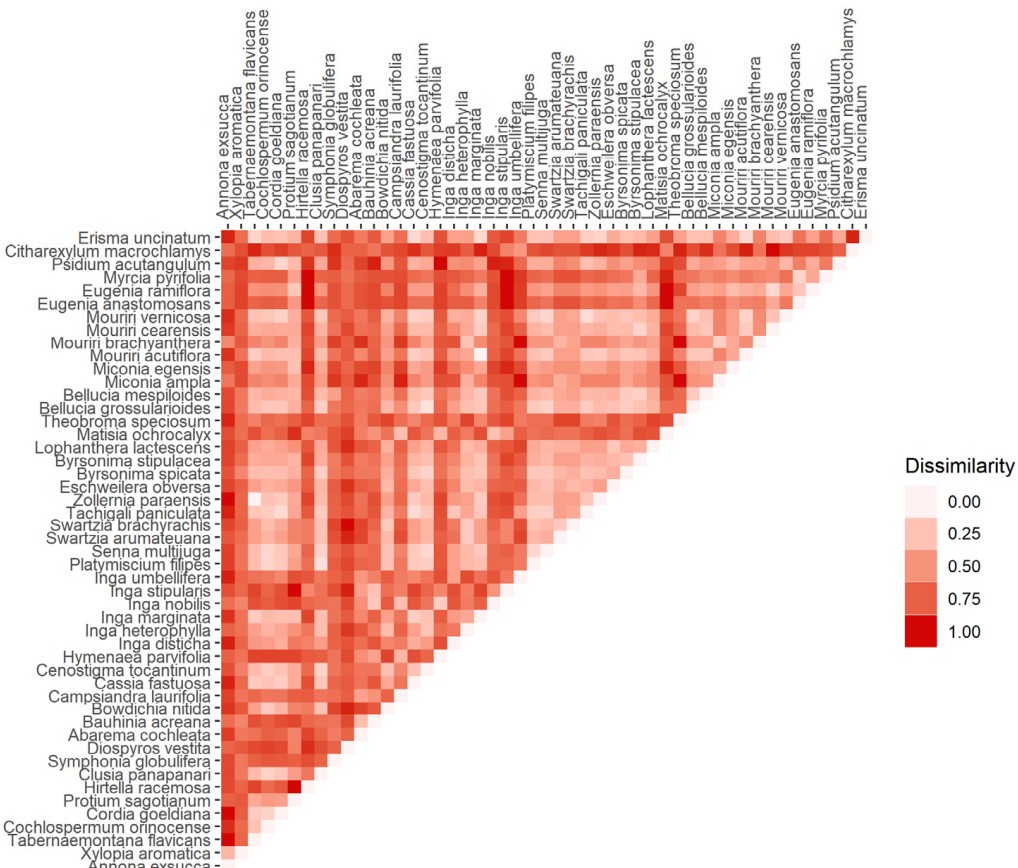

**Figure 4.** Dissimilarity between pairs of tree species, calculated using the generalized Gower distance regarding pollination syndromes, flower showiness and flowering period.

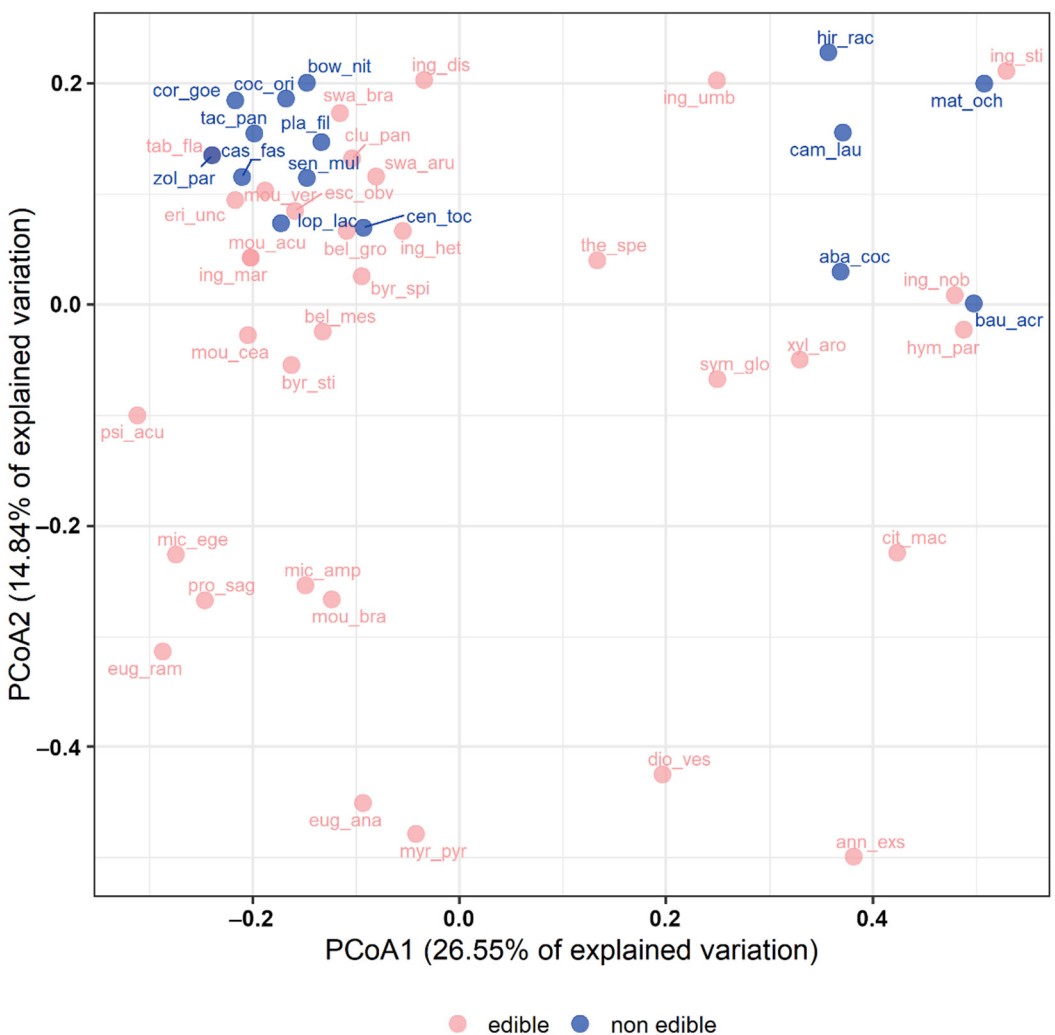

**Figure 5.** Principal Coordinate Analysis (PCoA) based on tree species dissimilarities calculated using the generalized Gower distance. Species are indicated by a 6-letter code (separated by "underline"), where the first three letters indicate the name of the genus and the last three letters the name of the species.

## 4. Discussion

Ornamental value was the criterion that excluded the largest number of the scored species (69.2%), was considered positive only for plants with showy flowers and flowering climaxes. We justify using this conspicuous flowering as this was often one of the main characters mentioned by the local population as desirable. The ornamental value fosters a connection between plants and people and is important if we are to tackle Aichi targets 1 and 2 [28]. Due to the relative shortage of wind-pollinated trees in the tropics, many of these species that have flowering climaxes also provide pollination resources for diverse groups of animals (for a breakdown of pollinator per species see Table 2).

The second criterion to exclude more species (52.5%). was their geographic distribution. Amazonian trees (in the Brazilian context) were selected over other trees to try to exclude plant stock from Southeastern Brazil that might not be as well adapted to the climate. Species that occur over a large geographic area are also more likely to be weedy or potentially invasive. During the scoring process we attempted to give preference to plants that were endemic to the state of Pará, or endemic to the Brazilian Amazon, however, upon analysing our data, the number of trees that fulfilled these rules was very low (only *Swartzia arumateuana*, *Aniba kappleri* Mez are endemic to Pará).

Considering the advantages of introducing species with fruits that are edible and encourage wildlife, the exclusion of 41.4% of species with dry, wind-dispersed or autochoric fruits seems to be a choice that does not need further explanation.

Excluding 30% of the species, height (up to 15 m tall) was a criterion that was used only for street plantings, and that could be waived if the plan was to use trees in open areas and parks, or even on the sides of wide roads.

Lack of any knowledge regarding propagation led us to negatively score only 24.3% of the species, mostly belonging to small, poorly known genera. We provide a list of these species to be used as an indicator of genera and species that need further work regarding seed germination and propagation, as information is almost absent from the literature (see Supplementary Material Table S2).

Resilience scored negatively for only 22.8% of the studied species. This is not surprising considering that the initial list [31] included plants not only from the ombrophilous forest but from other vegetation types in the area of Carajás. This geographical area can be said to be marginal in relation to the core Amazonian forest (Figure 1). We attempted to avoid suggesting trees that thrive strictly under very specific conditions (flooded *várzea* and *igapó* forest, understorey of *terra firme* forest) that might not succeed in more exposed situations and/or dry ground.

There are many other features that could be used for scoring urban afforestation species, such as perennial or deciduous foliage, density of shade, tolerance to pollution, drought, vulnerability to climate change and pests. However not enough is known about the Amazonian flora in order to be able to realistically score tree species at the moment, and all these aspects merit studies in the future.

Considering that seven species would not have been enough to offer all the benefits expected from urban afforestation, we provide a list of 49 species (Table 2) with further information regarding pollination syndromes (bees, butterflies, birds, bats) and phenology. This list may be used to select a cohort of species that encompasses the diversity of plant-animal interactions and/or, likewise, to choose a series of plants that flower and fruit at different times for their ornamental value and interest. Such selection may be guided to ensure there are resources to maintain wildlife, especially vertebrates, that depend upon resources throughout the year [61]. Having this in mind we provided a complementarity analysis (Figure 4, Supplementary Material Table S3) to enable practitioners to choose species pairs that are dissimilar in terms of their attributes to ensure that a maximum biodiversity is fostered by the plantings. Another way to guide managers is presented in the form of a PCoA analysis where the plants with edible and inedible fruits are displayed (Figure 5). This is another way to visualize and pick species that exhibit different features, such as flowering during different periods of the year or being pollinated by different agents.

The dominance of a few tree species in Amazonian urban areas [5] can lead to increased mortality of trees due to pest attacks. In addition, exotic species often perform less well than native species when temperature increases [62]. The lower vulnerability of native species to climate change also argues towards lower cost for maintenance of individual trees in urban areas. On the other hand, little knowledge regarding obtaining and growing seedlings of native trees hampers the plans of local governments, requiring the organization and involvement of rural producers to collect seed and provide young plants to the urban areas [63]. This occupation may bring benefits such as increasing the value of local forests, as trees come to be considered valuable sources of seeds and therefore of income for plant nurserymen. Moreover, as extreme events become more intense around the world, it is urgent to consider the native species presented here to ensure life quality in Amazonian urban afforestation projects.

Similar ranking to that presented here was used for degraded area recovery programmes in Amazonian Brazil [3], while a simpler version considering 104 trees was devised taking into account electricity lines for the state of Pará [36]. Even considering their hot, equatorial climate, Amazonian towns lack thoughtful approach and, country-wise, are among the least developed in terms of street, avenues or green area plantings [4]. Brazilian

southern and southeastern cities have approached urban greening in a more systematic and careful manner [64], but still exotic species predominate. The plan for Brasília, the country's capital, built in late 1950s, included detailed planning of tree planting from the onset, and these have now matured into a haven for wildlife and public enjoyment [11], even if the rapid growth of the capital has not included similar principles for its satellite cities. When considering the equatorial situation of the Amazon Basin, the challenges of urban tree-planting in its towns and cities may seem daunting. Reasons span from lack of horticultural knowledge of the region to low motivation from the authorities to promote public well-being through the provision and preservation of green areas. The city-state of Singapore has exemplified that it is possible to develop urban-greening to a fine standard, in a long term effort that involved five decades of investment [65], but that has only lately started to use species from Southeast Asia´s forests in its projects.

## 5. Conclusions

Floristic work, generating more authoritatively named tree checklists is needed to improve the knowledge of the Amazon flora and to safeguard its biodiversity. Such lists also provide important baselines for the selection and study of appropriate species for recovery of disturbed areas [3], as well as for rapidly growing cities and towns found within the biome.

Urban afforestation with local tree species goes towards meeting several of the Aichi targets [28] (Target 1, increase awareness of the value of biodiversity and its conservation; Target 2—to value biodiversity in local development; Target 8—pollution control, a target that may be met by planting more trees within cities; Target 14—preservation of ecosystem services and Target 15—climate change and restoration). The use of an initial list of trees with vouchers and consultation of databases and literature in order to understand the distribution and characteristics of the species suggested through this method also supports Target 9—to prevent introduction and establishment of invasive alien species.

Considering that municipalities are reportedly under strain to fulfill environmental duties in the area [30], it would be interesting to negotiate with the private sector possible solutions for the urgent afforestation of the fast-growing cities and towns in the Itacaiunas River basin. Local schools, agronomic institutes and university extension programmes could also contribute by tackling bottlenecks that are the local procurement and growing of seeds of suitable species.

**Supplementary Materials:** The following supporting information can be downloaded at: https: //doi.org/10.6084/m9.figshare.17910818, Table S1. Table with the complete dataset scored to select the species useful for urban afforestation. Table S2. Table highlighting the taxa that do not count with information regarding their propagation. Table S3. Matrix of dissimilarity between pairs of tree species.

**Author Contributions:** Conceptualization, D.C.Z., J.L.; Methodology, D.C.Z., J.L., A.H., C.O.A.; Formal Analysis, C.O.A., R.G.B.-S., F.M.; Data Curation C.O.A., R.G.B.-S., J.L., A.H., L.G.-S.; Writing— Original Draft Preparation D.C.Z.; Writing—Review and Editing, T.C.G., P.L.V., C.O.A., R.G.B.-S., D.C.Z., F.M., J.L.; Visualization, C.O.A., R.G.B.-S., F.M.; Project Administration, D.C.Z., T.C.G., P.L.V.; Funding Acquisition, D.C.Z., T.C.G. All authors have read and agreed to the published version of the manuscript.

**Funding:** The Capital Natural project counts with a grant from the National Council for Scientific and Technological Research (CNPq) (444280/2018-9) and DCZ holds a CNPq productivity grant (305301/2018-7).

**Informed Consent Statement:** Not applicable.

**Data Availability Statement:** Supplementary Materials (Table S1–S3) are available at: (10.6084/ m9.figshare.17910818).

**Acknowledgments:** We acknowledge the Museu Paraense Emílio Goeldi where the collections used for this work are housed, Vale for fieldwork support, Luiz Otávio Adão Teixeira for plant images, and Nigel P. Taylor has contributed with critical reading and English language improvement of this paper.

**Conflicts of Interest:** There are no conflict of interest to report.

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
