# Peer review of "Telling the Wood from the Trees: Ranking a Tree Species List to Aid Urban Afforestation in the Amazon"

_sustainability, doi:10.3390/su14031321_

Round 1

Reviewer 1 Report

My comments and suggestions for authors are included in the attached Word file.

Author Response

Dear reviewer 1

This is the letter from reviewer 1 with our answers in bold and italics

Dear authors,

Thank you very much for your work. In my opinion, the afforestation of urban areas with native trees is of great importance. Therefore I think your work is very necessary.

I think the manuscript is well structured and written. The use of species checklists for this purpose also reinforces the value of species catalogs, their applicability and rewards the hard work they involved. In addition, it is important to highlight the develpment of activities to achieve international targets. However, in my opinion, the manuscript is short-sighted and could cover more aspects to make it more interesting and useful. At the moment it contributes with a list of potential species to use in afforestation programms, but it could be more ambitious in terms of an ecological and management perspective. Therefore, I recommend major revisions.

Here, I include my comments per section. The lines refer to the Word document.

Abstract

I would add the method you used to select the species (Venn diagram) – ok, done

Line 40: species was further analysed – ok, corrected

Introduction

Line 82: While Brasília, the capital of Brazil, has an excellent example of urban planning that includes the marked presence of native and ornamental trees lining its streets and parks [11], the situation in a large number of cities in the Northeast [12–15] and North [16–18] regions of Brazil is much less encouraging.  – thanks, amended!

Line 91: Not only direct consequences for locals but also for the biodiversity, climate, ecological functions that, at the same time, have an indirect impact on the locals (microclimate, …). I would specify better in a sentence how these aspects indirectly benefit the locals. In addition, this part could be included in the paragraph starting in line 96 to put together all the benefits. – we have made the changes and expanded this part, however could not understand the reference to paragraph starting in line 96. Please clarify.

Line 102: International targets … Sentence too long. I would divide it in two. – done, thanks!

Line 140: Twenty-six  - ops… ok! Done it.

Line 142: three known to be toxic – done, thanks!

Line 156: If considering trees for urban parks and open areas, it is appropiate to use a list that does not take into account the height limit criterium. In this case, Hymenaea parvifolia (Fabaceae), Eschweilera obversa (Lecythidaceae) and Ersima uncinatum (Vochysiaceae) can be added to the list – done, thanks!

Line 161: Forty-nine

Material and Methods

I would change ecologic impact for ecological impact. – done, thanks!

I would explain why all the variables considered have the same weight and not one is considered more important than another. Very relevant, thanks. We added explanations to new lines 129-130. As we provided two types of dissimilarity analyses with weights, this was also explained in Methodology.

Results

Figure 1: The legend should be mor explicative. In the figure there is no A, B, C, … I would recommend including theses capital letters in the figure next to the variable each one refers to. Only the first part of the figure is addressed. It would be good to address also the second and third figures. The y axis of the second figure needs a title specifying what is measured (number of tree species). The third figure could be better explained: number of elements refers to number of species? The letters (a,b,c,d,e,f) were added to diagram A and graph B and the caption has been modified to include A diagram and B graph.

Table 2: To what does this (*) refer to? The asterisk (*) refers to the list of species that meet all six criteria. We added an explanation to the caption.

Portraits: I think the description of each species is not necessary (the first part). The flowering and fruiting time I would leave them in the text. As it is a list conceived to be used by managers, it would be better to include information that helps to decide which species is better for which place and the requirements for growing and maintenance. For example, it could include information about their habitat, ecological requirements for growing and maintenance (soil, water, …), information (if there is) about the root system, possible pests and stress conditions. – we are considering to leave the descriptions to help with the identification of the trees, but also included three new headers, Habitat and ecological requirements, Ecological role and Propagation, in order to provide the information suggested.

I would also recommend moving parts of the discussion in results, such as the first sentence of some paragraphs (e.g., line 250: The criterium that excludes the largest number of the scored species is the ornamental value (69.2%), line 259, line 268. These are results that should enrich the part of results. The discussion and explanation of these results (rest of the text) stays in the discussion section.  – ok, thanks! We summarized these results as a paragraph, and also amended the discussion so it did not become repetitive.

In addition, I would also give information about the ecological role of each species and about the presence of its pollinators in the region. I would include some references about this point and about the feedbacks. If there is information about interaction networks, that would be relevant.  – we included new header Ecological role to inform whether the tree species is a pioneer or climatic species, emergent or from the understorey. Unfortunately very little is known about the pollinators presence in the cities where the trees are to be introduced.

I would also include a recommendation of which species should be placed in which kind of areas of the cities (parks, avenues, streets, open spaces) and which species live well together (complementation of resources). Maybe in the form of a table, scheme, figure with the design of a city as a model, ... Maybe, you could do a case of study of a small city and take into account all the variables suggested (presence of pollinators, requirements, which zone for which species, combination of species, maintenance,…). Information about where to find or grow seedlings of these species (nurseries from the environmental offices, ...) would also be useful to enrich the applicability of this species list.  – Given the complexity of the data, we added two different dissimilarity analyses to aid practitioners to choose pairs of species and or species groups using plants that are different from each other. This implied in changes in methodology (new line 124 onwards), results (new line 335 onwards), adding two figures (4 and 5) and discussion (new lines 415 onwards). A new author from our ITV team, who helped with the analyses and critically read the paper, was also added.

Discussion

In my opinion, the discussion must be improved. I think it would improve the manuscript to put the research in a broader context. How has been done in other urban areas in the Amazon region? Have they considered the same variables? Compare with experiences in Brazil (e.g., Brasilia), in other countries or biomes. – we added a paragraph to provide context at the start of our discussion.

Also discuss the difficulties that may be encountered using these species. Also, the benefits in comparison to use alien species. – We continued the discussion at the end of the conclusion (new line 448 onwards) to include a more balanced discussion on these issues.

Wish you the best for your next version. Excellent! Here is the new version

Reviewer 2 Report

The article addresses the very important issue of selecting suitable native plants in the Amazon region, in human-transformed areas and especially in cities. The aim of the article is attractive in a practical and theoretical sense and it would certainly be good if it were presented in an international journal.

The article lacks a broader presentation of the geographical context of the region on which the authors base their research. I would suggest adding a figure with the location of the study area, indicating the towns and national parks that the authors refer to in the text. This is very important if the article was to serve other researchers. The Amazon area is vast and an appropriate spatial reference is necessary.

In the chapter "Materials and methods" the procedure is explained extremely briefly. The description of the chosen locators of the study area immediately turns into a very simplified description of the research method criteria used. I would particularly suggest a more extensive description of the "InteractiVenn" assumptions cited by the authors. Please explain why the criteria used were the best (?) What is meant by "adequate bibliography" - other researchers should have a chance to better understand which previous studies were taken into account.

From the point of view of the journal's profile, it seems important to refer to urban planning solutions that are currently not fulfilling their role in the study area. If the use of native plants in urban areas is to be widespread, how do the authors think this can be done under current political and economic conditions?

What is the purpose of the second abstract in Portuguese? Is it an editorial error or an intentional act, agreed with the journal's editorial board (?)

I would suggest additionally explaining better in the text:

1. Does the forest checklist only take place within conservation forms or does it apply to all forests in Brazil?

2.Please clarify whether the forest checklist is created only to protect areas of natural value, or whether it also has formal significance for land use planning?

Author Response

Dear Reviewer, many thanks for your careful reading of our paper. We are addressing your points one by one below (in bold and italics):

The article addresses the very important issue of selecting suitable native plants in the Amazon region, in human-transformed areas and especially in cities. The aim of the article is attractive in a practical and theoretical sense and it would certainly be good if it were presented in an international journal.

The article lacks a broader presentation of the geographical context of the region on which the authors base their research. I would suggest adding a figure with the location of the study area, indicating the towns and national parks that the authors refer to in the text. This is very important if the article was to serve other researchers. The Amazon area is vast and an appropriate spatial reference is necessary.

A map (Figure 1) has been prepared and included between Introduction and Material and Methods sections; We also added ref. 30 that gives a name to the area (Itacaiunas River Basin) to facilitate its location.

In the chapter "Materials and methods" the procedure is explained extremely briefly. The description of the chosen locators of the study area immediately turns into a very simplified description of the research method criteria used. I would particularly suggest a more extensive description of the "InteractiVenn" assumptions cited by the authors. Please explain why the criteria used were the best (?) What is meant by "adequate bibliography" - other researchers should have a chance to better understand which previous studies were taken into account.

This section was augmented to include a more detailed explanation of our procedures. The expression “relevant bibliography” was further explained and clarified. Unfortunately, apart from Flora of Brasil 2020 (ref 32) the bibliography for each Amazonian plant family and genus is very particular and, while some plant families are completely treated by revisions (Chrysobalanaceae, Meliaceae), others lack recent revisions or are seldom treated in local or regional floras, thus we relied mostly on online databases (ref 33 and 34) and also on own individual, field knowledge of the species.

From the point of view of the journal's profile, it seems important to refer to urban planning solutions that are currently not fulfilling their role in the study area. If the use of native plants in urban areas is to be widespread, how do the authors think this can be done under current political and economic conditions?

In the introduction we refer to Souza et al. 2015; Soares et al. 2021, Vieira et al. 2020 and dos Santos et al. 2021 (refs 4—7), all papers that look into urban planting in the Brazilian Amazon and show that the cities are not being planned adequately, not including enough tree planting and even less tree diversity. We now have added to the conclusion a paragraph exploring this situation in more detail, based on information from ref. 30.

What is the purpose of the second abstract in Portuguese? Is it an editorial error or an intentional act, agreed with the journal's editorial board (?)

We have now deleted the Portuguese abstract from the paper.

I would suggest additionally explaining better in the text:

  1. Does the forest checklist only take place within conservation forms or does it apply to all forests in Brazil?

The forest checklist we prepared (complete with voucher specimens deposited at herbaria MG and HCJS) is part of a local biodiversity study project undertaken by ITV and the Museu Paraense Emílio Goeldi. Because the variety of tree species in Brazil, this is still poorly understood in the Amazon (refs. 1 and 2), it is necessary to prepare local lists that are documented in order to obtain a real picture of the diversity in the region. We added an explanation making it clear that we used our local checklist to perform the tree species selection.

2.Please clarify whether the forest checklist is created only to protect areas of natural value, or whether it also has formal significance for land use planning?

A forest checklist is prepared initially to ascertain the biodiversity of protected (or any other, generally biodiverse) area, however it can be used for recovery of disturbed areas (Zappi et al. 2018, ref. 3), and for other initiatives, such as agroforestry and even sustainable logging. We highlighted this point in the first paragraph of the conclusion.

We hope this clarifies and attends all your requests. We will upload a file with the alterations into the system now.

We would like to warn you that reviews from reviewer 1 were extensive and meant that we added two dissimilarity analyses (figs 4 and 5), modifying abstract, methodology, results, discussion and bibliography. We also added a new author.

Round 2

Reviewer 1 Report

Dear authors,

Thank you for your new version. My comments can be found in the attached file.

Kind regards

Author Response

Dear Reviewer 1

Many thanks for spotting all these inconsistencies, we have now put them right. We are grateful for your contributions to our paper, that is much strengthened now.

Happy 2022

Zappi 

Reviewer 2 Report

The article has clearly been improved. I still think that the literature in the introduction and discussion chapter should have a broader context. Due to the regional nature of the paper, the current range of literature references can be accepted.

Author Response

Dear Reviewer 

We are extremely grateful for your contributions, which strengthened our paper very much. Below we exemplify 3 broader references used to provide a worldwide context of the subject we developed in our article, that, as you rightly said, is of a "regional" nature. Unfortunately, we could not find ways to further expand the introduction and discussion without escaping from our subject and hope you find it acceptable for the paper to progress.

All the best for 2022

Zappi and the team

Wiesel, P.G.; Dresch, E.; de Santana, E.R.R.; Loboestan, E.A. Urban Afforestation and Its Ecosystem Balance Contribution: A Bibliometric Review. Manag. Environ. Qual. Int. J. 2021, 32, 453–469, doi:10.1108/MEQ-07-2020-0156.

Beckmann-Wübbelt, A.; Fricke, A.; Sebesvari, Z.; Yakouchenkova, I.A.; Fröhlich, K.; Saha, S. High Public Appreciation for the Cultural Ecosystem Services of Urban and Peri‑urban Forests during the COVID-19 Pandemic. Sustain. Cities Soc. 2021, 74, 103240, doi:10.1016/j.scs.2021.103240.

Sirakaya, A.; Cliquet, A.; Harris, J. Ecosystem Services in Cities: Towards the International Legal Protection of Ecosystem Services in Urban Environments. Ecosyst. Serv. 2018, 29, 205–212, doi:10.1016/j.ecoser.2017.01.001.